# Defense and Offense Strategies: The Role of Aspartic Proteases in Plant–Pathogen Interactions

**DOI:** 10.3390/biology10020075

**Published:** 2021-01-21

**Authors:** Laura Figueiredo, Rita B. Santos, Andreia Figueiredo

**Affiliations:** Biosystems & Integrative Sciences Institute, Faculdade de Ciências, Universidade de Lisboa, Campo Grande, 1749-016 Lisboa, Portugal; lafigueiredo@fc.ul.pt (L.F.); aafigueiredo@fc.ul.pt (A.F.)

**Keywords:** proteases, development and reproduction, plant immunity, plant–pathogen interaction

## Abstract

**Simple Summary:**

Plants are sessile organisms that are continuously exposed to adverse environmental factors, both abiotic and biotic. Plant immunity is an intricate system that involves a remarkable array of structural, chemical, and protein-based layers of defense, aiming to stop pathogens before they cause irreversible damages. Proteases are an integral part of plant defense systems, with several hubs of action, from pathogen recognition and priming to the activation of plant hypersensitive response. Within this wide group of proteolytic enzymes, aspartic proteases have been implicated in several plant development functions and are gaining more prominence due to their involvement in plant–pathogen interactions. In this review, we summarize the current knowledge on plant and pathogenic aspartic proteases and highlight the most recent findings on their participation on plant defense, as well as in pathogen infection strategies.

**Abstract:**

Plant aspartic proteases (APs; E.C.3.4.23) are a group of proteolytic enzymes widely distributed among different species characterized by the conserved sequence Asp-Gly-Thr at the active site. With a broad spectrum of biological roles, plant APs are suggested to undergo functional specialization and to be crucial in developmental processes, such as in both biotic and abiotic stress responses. Over the last decade, an increasing number of publications highlighted the APs’ involvement in plant defense responses against a diversity of stresses. In contrast, few studies regarding pathogen-secreted APs and AP inhibitors have been published so far. In this review, we provide a comprehensive picture of aspartic proteases from plant and pathogenic origins, focusing on their relevance and participation in defense and offense strategies in plant–pathogen interactions.

## 1. The Past and the Present of Aspartic Proteases

Aspartic proteases (APs) were first discovered in animals during the nineteenth century. In 1836, Theodor Schwann described pepsin, which he identified during the study of animal gastric juices [1]. Later, in 1875, a pepsin-like proteinase was described in the pitcher plant (*Nepenthes*) after treatment of the plants’ digestive juice with sulfuric acid [2]. Almost a century after the discovery of the first AP, in 1930, the purification and crystallization of swine pepsin by John Northrop provided a substantial evidence that proteases were proteins [3]. In the following years, other proteases were crystallized and studied, including chymotrypsin, trypsin, and pepsinogen [4]. The conversion of pepsinogen to the active form of pepsin is an autocatalytic process that occurs at a low pH (1.5–5). Based on these findings, in 1962, the first step was taken towards the study of acidic proteinases [5]. In 1970, the discovery of pepstatin [6], a powerful inhibitor of aspartic proteases, encouraged its use as an immobilized compound for affinity purification of these proteases [7,8]. A major breakthrough occurred in 1972, when the complete amino acid sequence of the pig pepsin was uncovered [9]. Later, in the 1980s, the current terminology of the aspartic (or aspartyl) proteases was established, resulting from the observation that the carboxyl groups belonging to aspartate residues were involved in the catalytic process [10].

Although most studies about APs were performed in mammals, yeast, and fungi, some work has been developed in plants [11]. Plant APs were purified from the seeds of many organisms, such as *Oryza sativa* [12], *Cucurbita maxima* [13], *Cucumis sativus* [13], *Triticum aestivum* [14], and *Hordeum vulgare* [15], as well as from *Lycopersicon esculentum* leaves [16]. In 1991, the DNA sequence of the first plant aspartic protease, from barley (*Hordeurn vulgare*), was sequenced [17]. In the following years, APs from other organisms, including one from *Arabidopsis thaliana*, were isolated, providing more information about plant APs’ structure [7]. 

According to the MEROPS database (http://www.merops.ac.uk), aspartic proteases (EC 3.4.23) are grouped into 16 families, based on similarities of the amino acid sequences of the catalytic site. These families are clustered into five different clans that reflect a common evolutionary origin and similar tertiary structure [18]. Plant aspartic proteases are distributed among 12 of the 16 families: A1, A2, A3, A9, A11, A28, and A32 of clan AA; families A22 and A24 of clan AD; family A8 and A31 of clan AC and AE, respectively; and family A36 which has not yet been assigned to a clan [19]. A majority of plant APs belong to A1 family [20]. 

In 2004, with the completion of *Arabidopsis* genome, new perspectives have risen regarding plant APs’ diversity [21]. The first plant aspartic protease gene family to be described belonged to *Arabidopsis* with 51 known genes [22]. In the following years, plant APs have been found in increasing numbers [23] with 96 *OsAP* genes in rice (*Oryza sativa*) [24], 50 *VvAP* genes in grapevine (*Vitis vinifera*) [25], and 67 *PtAP* genes identified in black cottonwood (*Populus trichocarpa*) [26]. 

So far, it is known that plant APs are involved in several cell mechanisms, from developmental processes [27,28,29] to abiotic [30,31,32] and biotic stress responses [33,34,35]. Major milestones concerning aspartic proteases’ history and relation to pathogen resistance are presented in Figure 1.

## 2. The Features of Plant Aspartic Proteases

### 2.1. Structure and Classification

In the past years, with the study of phytepsin (AP from barley seeds) and cardosin (AP from the flowers of *Cynara cardunculus*), relevant information has been generated about plant aspartic proteases [22,36,37]. Plant APs, mostly belonging to family A1, are generally active at acid pH (pH 2–6), are specifically inhibited by pepstatin A, and comprise two aspartic acid residues essential for the catalytic activity [20,38]. The catalytic motifs of plant aspartic proteases from A1 family are usually Asp-Thr-Gly (DTG) or Asp-Ser-Gly (DSG) [20]. Although the general structure of the plant APs has similarities to that of mammals and microorganisms, plant APs contain a plant-specific insert (PSI) in the C-terminal region [7]. 

Most of the knowledge about A1 family plant APs comes from the study of typical APs, such as phytepsin and cardosin A and B [20]. Typical APs possess a signal peptide, a prosegment, and a PSI, and the catalytic site is composed by hydrophobic-hydrophobic-DTG-Ser-Ser residues (Figure 2). Exceptions to the structure of typical aspartic proteases were already described, as in the case of nucellin [39], in chloroplast nucleoid DNA-binding protein (CND41) [40], and in the constitutive disease resistance 1 (CDR1) protease [33]. These structural exceptions gave rise to three different categories, depending on the putative domain organization and active site sequence motifs: typical, nucellin-like, and atypical aspartic proteases [22]. Atypical and nucellin-like APs have distinct features on primary structure organization that differ from typical APs. The nucellin-like APs lack the prosegment and the PSI and comprise proteins similar to nucellin [20] with a characteristic sequence of residues: acidic-hydrophobic-DTG-serine-acidic residues around the catalytic site [22]. Atypical APs have intermediate features between typical and nucellin-like, and the active site is composed by hydrophobic-hydrophobic-DTG-Ser-acidic residues [20]. Both atypical and nucellin-like APs have a cysteine-rich region designated nepenthesin-type AP (NAP) specific insertion [38]. 

Detailed information on structure organization of plant aspartic proteases has been extensively reviewed in Reference [38].

### 2.2. Activation of Aspartic Proteases and Their Subcellular Localization

Proteolytic cleavage is crucial for active proteases. It starts with the removal of the signal sequence upon translocation to the ER lumen resulting in proproteins (zymogens). Usually, processing zymogens of typical plant APs involves the removal of the prosegment and partial or total deletion of PSI in an autocatalytic manner at the low pH of the vacuole [20,41]. Cheung and colleagues have proposed that, after proteolytic cleavage and activation, typical plant APs are either heterodimeric, where the PSI is partially digested (Figure 2a) [42,43,44] or entirely removed (Figure 2b; Reference [45]), or monomeric, without PSI (Figure 2c), as was observed in sweet potato SPAP1 [46]. 

There is evidence that two monomeric APs from potato tuber and leaves, StAsp1 [47] and StAsp3 [48], respectively, have the PSI in their mature form. However, the proteolytic mechanisms behind that process are still unknown [49,50]. In contrast, two atypical aspartic proteases, CDR1 and its rice homolog, have shown activity without the removal of the putative prosegment [51,52]. More studies have to be conducted to fully understand the inactivation mechanisms of plant APs. Soares and colleagues have recently proposed that the inactive form of APs occurs because the active site is blocked by the prosegment alone or by the prosegment together with the mature N-terminal and the flap. In contrast, precursors of cardosin A and B are active before undergoing the proteolytic process that removes prosegment, hence probably do not share the inactivation mechanism described above [20].

Considering the APs’ subcellular location, these proteases are found in various cellular compartments. Typical APs are mostly found in vacuoles, such as in the case of APs from barley [53], castor bean (*Ricinus communis*) [54], and *Arabidopsis* [55]. To a less extent, typical APs are also located in the extracellular space, such as in the case of tomato (*Solanum lycopersicum*) [16] and tobacco (*Nicotiana tabacum*) APs [56]. Atypical APs are widely distributed in the cell: *Arabidopsis* PCS1, ASPG1, and ASPR1 are located in the endoplasmic reticulum (ER) [29,30,57]; UNDEAD AP in mitochondria [58]; and CND41 and NANA in the chloroplast [27,40]. Rice OsAP65 is located in pre-vacuolar compartments [28]; nepenthesins and *Arabidopsis* AED1 and CDR1 are distributed in the extracellular space [33,59,60]. *Arabidopsis* A36 and A39 APs were found to be located in the plasma membrane as anchored proteins [61].

Plant APs are involved in many biological functions, particularly in developmental processes, such as chloroplast homeostasis and protein turnover [27,40], as well as in programmed cell death (PCD) and cell survival [62]. Developmentally controlled plant cell death is initiated through hormonal signaling, which in turn leads to the accumulation of reactive oxygen species (ROS) and transcriptional activation of PCD-related genes, such as proteases and nucleases. PCD can have different outcomes, such as senescence, the death of cells no longer required, or the creation of tissues that assume structural storage functions [63].

## 3. Aspartic Proteases Involved in Plant Defense Responses

Plants are sessile organisms, exposed to numerous biotic stresses and adverse environmental conditions [64]. Plant aspartic proteases were demonstrated to be involved in response to both abiotic [30,31,32] and biotic [33,34,35] environmental stressors.

### 3.1. Response to Abiotic Stress

Plant growth and productivity have been impaired due to abiotic stresses, such as drought, heat, cold, and excess of salt and metals in the soil [65]. Under drought stress, an aspartic protease from common bean (*Phaseolus vulgaris*), *PvAP1*, was shown to be up-regulated earlier in the leaves of a drought-susceptible cultivar than in the resistant cultivar (Table 1; Reference [66]). The *Arabidopsis* aspartic protease in guard cell 1 (ASPG1) was firstly shown to be involved in drought stress resistance, in addition to its role in the degradation of seed storage proteins [30]. *Arabidopsis* mutants overexpressing *ASPG1* were shown to recover more efficiently from drought, as ASPG1 lead to a significant increase in abscisic acid (ABA) sensitivity by guard cells and antioxidant enzymes activation, preventing *Arabidopsis* plants from oxidative damage [30]. A gene homologous to *ASPG1* from potato was shown to be down-regulated under drought and up-regulated upon re-watering, suggesting also a role in drought stress (Table 1; Reference [67]).

Recently, an *Arabidopsis* aspartic protease, APA1, has been implicated also in drought tolerance. Plants overexpressing the *apa1* gene (OE-*APA1*) were more tolerant to mild water deficit (MWD) than WT plants, while *apa1* line was more susceptible (Table 1; Reference [31]). OE-*APA1* lines exhibited more total leaf area, less chlorophyll content and shortened principal root length under MWD treatment. Analysis of stomatal behavior showed that OE-*APA1* plants presented reduced stomatal pore aperture and reduced stomatal index [31]. Since ABA regulates stomatal closure, aperture pore size was determined upon ABA treatment. The stomata of OE-*APA1* plants was already closed before ABA treatment. This work suggested that *APA1* has a role in stomatal behavior via regulation of the ABA signaling pathway [31]. An aspartic protease gene from buckwheat (*Fagopyrum esculentum*), *FeAP9*, was found to be up-regulated in leaves in response to numerous abiotic stresses, including dark, drought, wounding, and UV-B light (Table 1; Reference [69]). Moreover, when pineapple fruit (*Ananas comosus*) is exposed to a chilling injury, it develops brown symptoms known as blackheart [70]. Gene expression analysis of the pineapple fruit under postharvest chilling treatment showed that aspartic protease *AcAP1* was up-regulated in a variety resistant to blackheart and down-regulated on a susceptible one (Table 1; Reference [70]). Thus, it is expected that AcAP1 is involved in resistant mechanisms concerning chilling stress [70]. Transgenic *Arabidopsis* plants overexpressing the grapevine aspartic protease *AP17* showed salt- and drought-tolerance as transgenic seeds had higher levels of germination and transgenic seedlings roots were longer under osmotic stress [32]. In addition, the plasma membranes of the transgenic seedlings suffered less damage, and the genes involved in ABA biosynthesis were up-regulated (Table 1; Reference [32]). These results suggest that AP17 is a key component for maintaining the integrity of the membrane and may be involved in ABA biosynthetic pathway [32].

Heavy metals, such as iron, copper, nickel, mercury, and cadmium, are one of the environmental pollutants affecting plant growth. Despite the fact that plants have the ability to tolerate certain concentrations of these metals, when a certain level is exceeded, it causes toxicity, leading to the generation of ROS [71]. Although plant extracts have been reported to prevent heavy-metal-induced stress [72], only cysteine proteases, such as caspases and vacuolar processing enzymes, have been shown to be involved in this process [73,74,75].

### 3.2. Aspartic Proteases Involvement in Plant–Pathogen Interaction

The first clue of the involvement of plant APs in biotic stress was found in tomato leaves, where pathogenesis-related (PR) proteins secreted upon pathogen challenge were degraded by an extracellular aspartic protease, preventing its over accumulation [16]. Three years later, the same function was proposed for an aspartic protease found in tobacco leaves [56]. The tomato aspartyl protease is thought to be responsible for the cleavage of PR-1b [76]. PR-1b cleavage releases a peptide, CAP-derived peptide 1 (CAPE1), that induces the expression of genes involved in stress and defense responses, innate immunity, and systemic acquired resistance (SAR) [77]. It was suggested that CAPE1 may act as a novel damage-associated molecular patterns (DAMP) linked to both jasmonic acid (JA) and salicylic acid (SA) pathways and SAR activation [77]. An aspartic protease gene detected in tomato leaves (*LeAspP*), in response to wounding and treatments with systemin and methyl jasmonate (MeJA), was also shown to be systemically induced, suggesting that this AP plays a role in defense against pathogens [78]. In potato tubers, the *Solanum tuberosum* aspartic protease 1 (StAP1) was identified in immunological analysis of intercellular washing fluids of potato tubers has presenting a higher accumulation in a resistant cultivar than in a susceptible one upon *Phytophthora infestans* infection [79]. Moreover, western blot analysis of an AP from potato leaves, *Solanum tuberosum* aspartic protease 3 (StAP3), showed a higher accumulation in potato resistant cultivar upon *P. infestans* infection [80]. StAsp, also isolated from leaves, presented a higher expression in the resistant cultivar after infection with *P. infestans* [49]. These results suggest that several StAPs may be involved in plant immunity response. In a recent study focusing on the soybean–*Phytophthora sojae* interaction, a secreted soybean AP (GmAP5) has been described to bind and degrade the pathogen effector PsXEG1, an apoplastic endoglucanase [81]. The cleavage of this effector severely affects *P. sojae* virulence (Figure 3a). Soybean has another layer of defense towards this *P. sojae* effector. The apoplastic inhibitor protein GmGIP1 bind to the PsXEG1 effector, reducing its enzymatic activity and, thus, the pathogen’s virulence. To counterattack this, *P. sojae* N-glycosilates this effector, protecting it from the proteolytic activity of GmAP5, and secretes a decoy effector, PsXLP1, that binds more tightly to GmGIP1 [81,82]. This intricate system shows the different layers of defense and offense in plant–pathogen interactions.

More aspartic proteases were described to induce systemic defense responses in plants. The *Arabidopsis CDR1* gene was identified while studying a gain-of-function dominant mutation that presented a phenotype of enhanced resistance to the bacterial pathogen *Pseudomonas syringae* [33]. *Arabidopsis CDR1* mutants exhibited a phenotype that mimics constitutive activation of SAR, including the accumulation of high levels of SA; SAR transcripts marker genes, such as *PR1* and *PR2*; and oxidative bursts resulting from hypersensitive response (HR). It was hypothesized that CDR1 released a peptide elicitor that may function as a mobile SAR signal [33]. Ectopic expression of the rice ortholog *CDR1* (*OsCDR1*), in both *Arabidopsis* and rice, conferred enhanced resistance to *Pseudomonas syringae*, *Hyaloperonospora arabidopsidis*, *Xanthomonas oryzae*, and *Magnaporthe oryzae*, which is correlated with the enhanced PR gene expression [83]. Infiltration of *Arabidopsis* leaves with purified OsCDR1–GST fusion protein induced *PR2* expression. Interestingly, the expression of this pathogen related protein was also identified in non-inoculated neighbor leaves, demonstrating that OsCDR1 induces systemic defense [52]. Conversely, apoplastic enhanced disease susceptibility 1 aspartic protease (AED1) has been described as having a role in SAR repression [59]. In enhanced disease susceptibility 1 (*eds1*) mutant plants, which are SAR defected, AED1 was found in a proteome profiling analysis of the extracellular fluid in response to the *P. syringae* effector AvrRpm1 [59]. The AED1 transcript accumulation was found to be induced by this pathogen both locally and systemically in WT and *eds1* mutants, despite AED1 content was much lower in the *eds1* background [59]. These results suggest that systemic accumulation of AED1 in response to *P. syringae* depends on EDS1 protein and, thus, is correlated with SAR [59]. Additionally, overexpression of *AED1* led to the repression of both SAR- and SA-induced resistance without affecting *P. syringae* growth in healthy plants. These findings support the hypothesis that AED1 might be part of a homeostatic mechanism to limit SAR signaling and to reallocate resources from defense to plant growth [59]. Another recent report has highlighted the importance of aspartic proteases in plant defense by demonstrating its antibacterial function. Wang and colleagues showed that *A. thaliana* secreted aspartic protease 1 and 2 (SAP1 and SAP2) are able to cleave a highly conserved bacterial protein, MucD. The cleavage of MucD inhibits the growth of *P. syringae in planta* and *in vitro*, showing the importance of antibacterial mechanisms in plant defense [84]. 

The activity of the rice aspartic protease 77 gene (*OsAP77*) in rice transgenic lines was induced upon infection by *M. oryzae*, *X. oryzae*, and cucumber mosaic virus (CMV) in vascular tissues [34]. In addition, rice transgenic plants treated with SA, isonicotinic acid, hydrogen peroxide, and ABA showed an increased level of the reporter gene GUS. These results suggest that OsAP77 has a positive role in pathogens defense [34]. Moreover, two APs were shown to be induced in the rice apoplast upon *M. oryzae* infection, suggesting that these proteins may act as signal transductors apart from their hydrolytic activity [85]. The Bcl-2 associated athanogene protein 6 (BAG6) participates in limiting pathogen colonization and spread of the necrotrophic fungus *Botritis cinerea* by inducing autophagy [86]. For autophagy to occur, full-length BAG6 needs to be activated by protease processing. Recently, Arabidopsis aspartyl protease cleaving BAG (APCB1) was demonstrated to be essential in BAG6 proteolytic processing [87]. The *apcb1* mutants exhibited enhanced susceptibility similar to the *bag6* mutants. Mutation of the catalytic site of APCB1 led to the absence of BAG6 cleavage. This aspartic protease appears to be BAG6 specific, once the two unrelated *Arabidopsis* aspartyl proteases were unable to cleave BAG6 in *apcb1* mutant line, causing loss of resistance [87].

Previous studies have also shown that the grapevine aspartic protease 13 (*AP13*) gene was up-regulated in Chinese wild *Vitis quinquangularis* cv. “Shang-24” following *Erysiphe necator* infection. It was also up-regulated in *V. labrusca* × *V. vinifera* cv. “Kyoho” following a treatment with SA, suggesting that this gene may confer resistance to biotrophic pathogens [25]. Further studies analyzed the expression levels of *AP13* upon inoculation with *B. cinerea* and treatments with hormones involved in plant defense. *AP13* from *V. quinquangularis* cv. “Shang-24” was shown to be up-regulated after both SA and ethylene (ET) treatment and was down-regulated upon JA and MeJA treatment and *B. cinerea* infection. These results suggest that AP13 promotes the SA dependent signal transduction pathway and suppresses the JA signal transduction pathway. The ectopic expression of *AP13* in *Arabidopsis* improved the resistance of transgenic plants to *E. necator* and the bacterial pathogen *P. syringae* but reduced the resistance to *B. cinerea* [35].

The perception of microbial or damage signals by plants’ receptors initiates a response that leads to the production of peptides and small molecules that enhance immunity responses [88]. Several studies have highlighted the antimicrobial activity of plant proteases. Potato-isolated APs were shown to inhibit both *Phytophthora infestans* and *Fusarium solani* growth; salpicornin, isolated from *Salpichroa origanifolia* fruits, was shown to inhibit *Fusarium solani* growth; and cirsin, isolated from thistle plants also presented antifungal activity against *Lewia infectoria*, *Alternaria alternata*, and *Drechslera biseptata* [79,80,89,90].

Monomeric aspartic proteases from *S. tuberosum* were reported to have bifunctional activity, both proteolytic and antimicrobial [91]. The plant-specific insert of *S. tuberosum* aspartic protease 1 (StAsp-PSI) is able to interact with pathogens’ spore surface, inducing damage in its plasma membranes and causing death in a dose-dependent manner [91]. *In vitro*, StAsp-PSI was able to kill spores of *P. infestans* and *Fusarium solani* by direct interaction of the protein with the pathogens’ cell membrane, leading to an increased permeability and lysis [91]. A recent study has suggested that the PSI from *S. tuberosum* facilitates membrane fusion at acidic pH, while the mature AP degrades pathogenic proteins in the extracellular space [92]. *Arabidopsis* transgenic lines expressing the potato PSI increased plant resistance to *B. cinerea* infection through direct cytotoxic activity and induction of gene expression associated to the regulation of JA and SA pathways, such as *PDF1.2* and *PR-1* [50]. Moreover, cytotoxic analysis of salpichroin and circin activity suggested it was not related to proteolysis but to membrane permeabilization of pathogen conidia. Altogether, these studies suggest that the PSI domain may be involved in membrane permeabilization [89]. Therefore, the constitutive expression of these APs’ PSI could potentially be used as a strategy to cope with plant pathogens [50].

## 4. Pathogen Aspartic Proteases

Proteases from pathogens can also play an important role in the mechanisms of virulence during infection, by participating in the degradation of the host’s physical barriers and combating the host’s defense mechanisms [93]. However, aspartic proteases’ role in virulence of plant pathogens is still poorly studied [93,94,95]. 

*Botrytis cinerea*, a necrotrophic fungus that causes the grey mold disease in many plant species, has a significant content of secreted APs upon successful infection of host [96]. In 1990, APs were first established as possible *B. cinerea* virulence factors when the supplementation of inoculum with pepstain A (AP inhibitor) limited infection [97]. *BcAp5*, *BcAP8*, *BcAP9* and *BcAP14* were observed to be up-regulated in the first hours of infection upon grape berry infection [98]. A recent report has also shown that, in the first 24 hours of infection, there is an increase in the transcription of phytotoxins and cell wall degrading enzymes [99]. Taken together, these studies show that APs are a crucial tool for the *B. cinerea* infection strategy.

A genome analysis of three *Phytophthora* species (*P. infestans*, *P. sojae*, and *P. ramorum*) reported, in 2011, that these pathogens present 5 clans and 12 families of APs and that they are all predicted to be membrane-bound [100]. APs have also been identified in the secretome of four *Phytophthora* species (*P. infestans* [101] and *P. pseudosyringae*—forest pathogen; *P. chamydospora* and *P. gonapodyides*—frequent in aquatic habitats; Reference [102]). In a recent study, *P. infestans* (Pi) transformants, overexpressor, and silenced lines for APs, were characterized to determine if *P. infestans* aspartic proteases (PiAPs) play a role in virulence. *PiAP10* and *PiAP12* silenced lines showed a reduction in mycelial growth and sporangia production and low infection efficiency on inoculated potato leaves. Activity assays suggest that both lines were capable to cleave the *P. infestans* Arginine-x-Leucine-Arginine effector AVR4. *PiAP11* silenced transformants did not show any reduction. These results suggest that *PiAP10* and *PiAP12* play a role in virulence (Figure 3b) [103]. An effector of *P. sojae* (PsAvh240) has been recently described to interact with an AP (GmAP1) from a resistant soybean cultivar (Figure 3a). This interaction suppresses GmAP1 secretion to the apoplast, limiting soybean apoplastic immunity and, thus, plant defense [104]. 

The infection mechanism of *Fusarium proliferatum*, a pathogen that causes fungal keratitis in several crops [105], has been shown to be pH-dependent [106]. At an alkaline pH, the infection of bananas by *F. proliferatum* is hampered. At the same time, in these conditions, a secreted AP from this pathogen is down-regulated. Although further studies are needed, the authors suggest that this AP may be crucial for the infection process of *F. proliferatum* [106]. A wheat fungal pathogen, *Zymoseptoria tritici*, has been described to up regulate extracellular APs during its asymptomatic biotrophic phase [107]. Further studies could confirm that these APs may be effector genes that suppress wheat apoplastic immunity. During infection of sunflower cotyledons by *Sclerotinia sclerotiorum,* a necrotrophic fungus, several acid proteases were shown to be secreted and the *aspS* gene, encoding for an aspartic protease, was expressed at the early stages of infection [108]. Studies have shown that an acid AP from *Fusarium culmorum* presented a role in the pathogen infection due to its capacity to degrade plant inhibitor proteins such as bean polygalacturonase inhibitor and soybean trypsin inhibitor [109]. Overexpression of *endothiapepsin* (*Epn*), an AP secreted by the *Cryphonectria parasitica* fungus responsible for chestnut blight, leads to enhanced necrosis on chestnut bark and wood tissues, suggesting its involvement in pathogenicity, as well [110]. The extracellular aspartic proteases Eap1 from *Sporisorium reilianum* (that causes maize and sorghum head smut) and APSm1 from *Stenocarpella maydis* (that causes diplodia ear and stalk rot in maize) were purified and characterized [111,112]. Strong evidences highlight their involvement as key components of the biological and infection cycles of the pathogen [111,112]. *Ralstonia solenacearum*, a pathogenic bacterium causing bacterial wilt on many solenaceous crops, secretes an AP (Rsa1) that is able to elicit HR response in potato and has an important role in the virulence of this pathogen [113]. The proteases of the biotrophic pathogen *Cladosporium fulvum*, which causes tomato leaf mold disease, were analyzed in a recent study, by transcriptome and proteome analysis. In total, 14 out of the 59 predicted protease genes were expressed *in vitro* and *in planta*, and one of proteins (CfPro7) was predicted to be secreted [114]. During the infection of apples by *Penicillium expansum*, an AP (*PEX2_009280*) was reported to be up-regulated, suggesting its possible role in this pathogen infection mechanism [115]. Another apple fruit pathogen, *Colletotricum acutatum*, has been shown to secrete an AP during infection. A protease inhibitor extracted from apple fruits, that showed a similar activity to a commercial AP inhibitor, inhibited *C. acucatum* growth *in vivo* and *in vitro*, showing the importance of this AP in the virulence of this pathogen [116].

The studies reviewed here emphasize the diversity of pathogens that rely on the proteolytic activity of aspartic proteases for the success of their infection process. However, further studies are needed to deepen our knowledge on the role of proteases, as well as protease inhibitors, as virulence factors.

## 5. Conclusions and Future Perspectives

In the past years, several studies regarding the role of plant aspartic proteases in different cell functions have been reported. Although there is a lot of information about plant typical APs, there is still a need for a better understanding of the structure of atypical and nucellin-like APs, as well as their substrates, interacting proteins, and proteolytic activity. Since the end of the 19th century, the study of the involvement of APs in plant development has increased. Although many of the functions given to APs remain hypothetical, studies using reverse engineering tools and biochemical studies are essential to understand the detail biological function. The knowledge about plant–pathogen interaction has risen, as well as the involvement of aspartic proteases in this interaction. It is clear that APs have an important role in plant defense against a wide range of pathogens. On the other hand, aspartic proteases from pathogens are still poorly studied. Research within a variety of molecular, genetic, and biochemical approaches will contribute to fully address these questions and finally understand the regulation mechanisms regarding plant aspartic proteases role, particularly in plant development and pathogen interaction.

## Figures and Tables

**Figure 1 biology-10-00075-f001:**
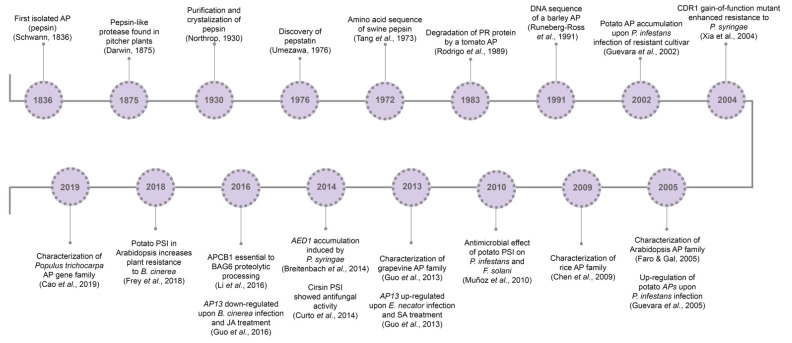
Aspartic proteases historical breakthroughs. AP, aspartic protease; CDR1, constitutive disease resistance 1; PSI, plant-specific insert; AP13, aspartic protease 13; SA, salicylic acid; AED1, apoplastic enhanced disease susceptibility 1; APCB1, aspartyl protease cleaving bcl-2 associated athanogene; BAG6, BCL-2 associated athanogene protein 6; SA, salicylic acid; JA, jasmonic acid; MeJA, methyl jasmonate.

**Figure 2 biology-10-00075-f002:**
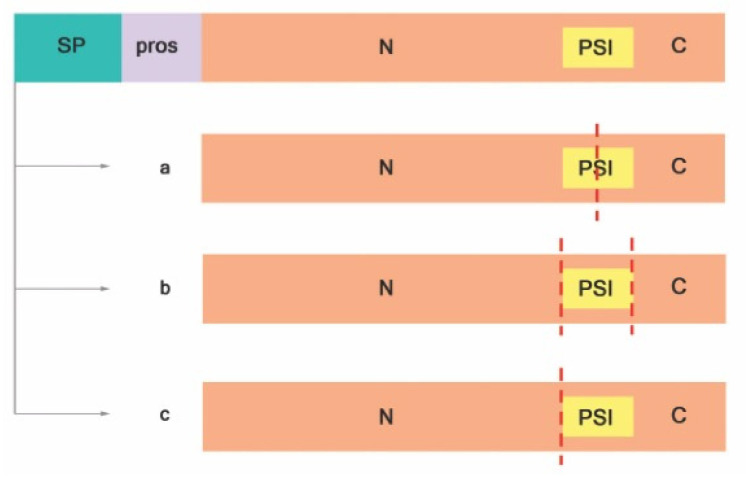
Proteolytic activation of typical APs adapted from Reference [41]. (**a**) PSI is digested at the midsection, (**b**) PSI is entirely removed, and (**c**) PSI and C-terminal are removed. Signal peptide (SP); prosegment (pros); plant-specific insert (PSI); N-terminal domain (N); C-terminal domain (C); red dashed lines indicate cleavage sites.

**Figure 3 biology-10-00075-f003:**
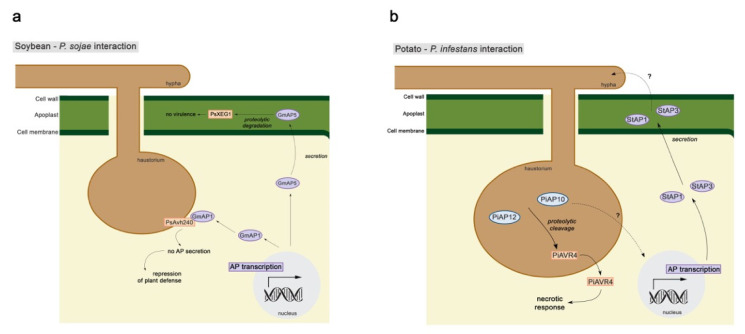
Schematic representation of (**a**) soybean–*P. sojae* and (**b**) potato–*P. infestans* interactions. Purple ellipses, plant aspartic proteases; orange boxes, pathogen effectors; dashed arrows, unknown processes.

**Table 1 biology-10-00075-t001:** Aspartic proteases involved in abiotic stress responses.

Plant	Protease	Role	Reference
*Phaseolus vulgaris*	PvAP1	Up-regulated in susceptible cultivars	[66]
*Arabidopsis thaliana*	ASPG1	Overexpressor lines recover better from drought	[30]
*Solanum tuberosum*	ASPG1 homolog	Down-regulated in drought and up-regulated upon rewatering	[68]
*Arabidopsis thaliana*	APA1	Overexpressor lines are more tolerant to MWD	[31]
*Fagopyrum esculentum*	FeAP9	Up-regulated in dark, drought, UV-B light and wounding stresses	[69]
*Ananas comosus*	AcAP1	Up-regulated upon chilling treatment in resistant to chill cultivars	[70]
*Vitis vinifera*	VvAP17	Expression in Arabidopsis increased tolerance to drought and salt stress	[32]

MWD, mild water deficit.

## Data Availability

No new data were created or analyzed in this study. Data sharing is not applicable to this article.

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
