# Peer review of "Defense and Offense Strategies: The Role of Aspartic Proteases in Plant–Pathogen Interactions"

_biology, 2021, doi:10.3390/biology10020075_

Round 1
Reviewer 1 Report
The eview on the role of aspartic proteases in plant-pathogen interactions sums up once again the knowledge on plant and pathogenic aspartic proteases and their participation on plant defense. Unfortunately the authors do not make convincingly clear, in which way this summary helps to understand the role of aspartic proteases in plant-pathogen interactions as compared to already published reviews. This way it seems of low interest to the general reader of this journal.
Author Response
Reviewer 1
Reviewer comment: The review on the role of aspartic proteases in plant-pathogen interactions sums up once again the knowledge on plant and pathogenic aspartic proteases and their participation on plant defense. Unfortunately the authors do not make convincingly clear, in which way this summary helps to understand the role of aspartic proteases in plant-pathogen interactions as compared to already published reviews. This way it seems of low interest to the general reader of this journal.
Response to reviewer: Our review summarizes the latest knowledge on both plant and pathogenic aspartic proteases, including novel insights published in 2018-2020. To our knowledge, in the last five years, the most relevant-to-the-field publications have mainly addressed plant aspartic proteases and did not present a dedicated subchapter to pathogenic proteases (please see some examples below). The information on the pathogen side of the interaction is also relevant and has been overlooked in plant-pathogen interactions reviews. Thus, we think that this review is extremely relevant to the filed, since it gathers the latest information available on aspartic protease in the both sides of the plant-pathogen interaction.
Soares A, Ribeiro Carlton SM, Simões I. Atypical and nucellin-like aspartic proteases: emerging players in plant developmental processes and stress responses. Journal of Experimental Botany. 2019;70:2059-2076. doi: 10.1093/jxb/erz034.
Hou, S.; Jamieson, P.; He, P. The cloak, dagger, and shield: proteases in plant–pathogen interactions. Biochem. J. 2018, 475, 2491–2509, doi:10.1042/BCJ20170781.
Thomas, E.L.; Van der Hoorn, R.A.L. Ten Prominent Host Proteases in Plant-Pathogen Interactions. Int. J. Mol. Sci. 2018, 19, 639. doi:10.3390/ijms19020639
Mukhi, N., Gorenkin, D. and Banfield, M.J. (2020), Exploring folds, evolution and host interactions: understanding effector structure/function in disease and immunity. New Phytol, 227: 326-333. doi:10.1111/nph.16563

Reviewer 2 Report
The Review entitled: Defense and Offense Strategies: the Role of Aspartic 2 Proteases in Plant-Pathogen Interactions summarize the current knowledge on plant and pathogenic aspartic proteases, highlight the most recent findings on their participation on plant defense as well as in pathogen infection strategies.
The review is well written but in my opinion it lacks some aspects. It should be integrated considering the following aspects:
In the introduction describe the application of Aspartic Proteases in Food Industry because their applications are well established in the processing and manufacturing of both traditional and novel food products. There are some recent papers on Plant Aspartic Proteases for Industrial Applications
Plant proteases play a fundamental role in several processes like growth, development and in response to biotic and abiotic stress. In particular, aspartic proteases are expressed in different plant organs and have antimicrobial activity. Extend this aspect comparing the antimicrobial activity of the known aspartic proteases and that of proteins extracted from some fruit. In addition, there are works which report that potential antioxidant peptides are produced by using aspartic proteases . Extend this aspect in the introduction or in discussion and for this aim I suggest to read and quote the following paper: Antimicrobial and antioxidant activity of proteins from Feijoa sellowiana Berg. fruit before and after in vitro gastrointestinal digestion. Nat Prod Res. 2020 Sep;34(18):2607-2611. doi: 10.1080/14786419.2018.1543686.
Proteases are crucial for living cells and play a role in plant cell adaptation to environmental conditions. Oxidative stress produced oxidized proteins which are selectively degraded by proteases. To understand the role of proteolysis in response to metal stress there are many studies that indicate that different proteases are involved in plant defence against metal toxicity. Some fruit extract are able to protect some cells from Mercury-Induced Cellular Toxicity. Is there any correlation between protease and metal defense? I suggest to read and quote the following paper: Phenol-Rich Feijoa sellowiana (Pineapple Guava) Extracts Protect Human Red Blood Cells from Mercury-Induced Cellular Toxicity. Antioxidants (Basel). 2019 Jul 11;8(7):220. doi: 10.3390/antiox8070220
What is known about the inhibition of these proteases by metals?
What is known about the Plant proteases and programmed cell death?
Gender and species names must be written in italics, some examples: line 208 -209 (correct in all the manuscript).
Improve image quality and preferably choose colors that match each other.
Author Response
Reviewer 2
The Review entitled: Defense and Offense Strategies: the Role of Aspartic 2 Proteases in Plant-Pathogen Interactions summarize the current knowledge on plant and pathogenic aspartic proteases, highlight the most recent findings on their participation on plant defense as well as in pathogen infection strategies.
The review is well written but in my opinion it lacks some aspects. It should be integrated considering the following aspects:
Reviewer comment: In the introduction describe the application of Aspartic Proteases in Food Industry because their applications are well established in the processing and manufacturing of both traditional and novel food products. There are some recent papers on Plant Aspartic Proteases for Industrial Applications
Response to reviewer: The authors acknowledge the reviewer suggestion. This review focuses on plant and pathogen aspartic proteases and their role in several processes on plant-pathogen interaction. Our introduction describes briefly the history of aspartic proteases discovery and we believe that including information on plant aspartic proteases for industrial application would be out of the scope of this review. Moreover, it was recently published (Folgado et al. 2020; 10.3390/plants9020147) a review entitled “Plant Aspartic Proteases for Industrial Applications: Thistle Get Better” covering the most updated research on that subject.
Reviewer comment: Plant proteases play a fundamental role in several processes like growth, development and in response to biotic and abiotic stress. In particular, aspartic proteases are expressed in different plant organs and have antimicrobial activity. Extend this aspect comparing the antimicrobial activity of the known aspartic proteases and that of proteins extracted from some fruit. In addition, there are works which report that potential antioxidant peptides are produced by using aspartic proteases . Extend this aspect in the introduction or in discussion and for this aim I suggest to read and quote the following paper: Antimicrobial and antioxidant activity of proteins from Feijoa sellowiana Berg. fruit before and after in vitro gastrointestinal digestion. Nat Prod Res. 2020 Sep;34(18):2607-2611. doi: 10.1080/14786419.2018.1543686.
Response to reviewer: The authors acknowledge reviewer suggestion; in fact, plant aspartic proteases were shown to be involved in several processes from development to stress response. The authors agree that mentioning antimicrobial activity of aspartic proteases is important and a paragraph was already included in the original version. However, in order to clarify this info for the readers, we have altered this paragraph and included more information (line 273-292). Considering the layout of the manuscript, the authors only focused on the aspartic proteases with antimicrobial activity and not on fruit extracts.
Reviewer comment: Proteases are crucial for living cells and play a role in plant cell adaptation to environmental conditions. Oxidative stress produced oxidized proteins which are selectively degraded by proteases. To understand the role of proteolysis in response to metal stress there are many studies that indicate that different proteases are involved in plant defence against metal toxicity. Some fruit extract are able to protect some cells from Mercury-Induced Cellular Toxicity. Is there any correlation between protease and metal defense? I suggest to read and quote the following paper: Phenol-Rich Feijoa sellowiana (Pineapple Guava) Extracts Protect Human Red Blood Cells from Mercury-Induced Cellular Toxicity. Antioxidants (Basel). 2019 Jul 11;8(7):220. doi: 10.3390/antiox8070220. What is known about the inhibition of these proteases by metals?
Response to reviewer: The role of proteases in heavy metal stress is very interesting. We have added a paragraph to the manuscript has suggested by the reviewer; please see lines 179-186.
Reviewer comment: What is known about the Plant proteases and programmed cell death?
Response to reviewer: A paragraph on APs and programmed cell death was added to the manuscript; please see lines 135-141.
Reviewer comment: Gender and species names must be written in italics, some examples: line 208 -209 (correct in all the manuscript).
Response to reviewer: This has been corrected, throughout the manuscript, as suggested by the reviewer.
Reviewer comment: Improve image quality and preferably choose colors that match each other.
Response to reviewer: The quality of the images uploaded in the platform is of 300 dpi; images have been improved as suggested by the reviewer.

Round 2
Reviewer 1 Report
The authors cite four publications as proof, that their review would contain novel information with latest results. Unfortunately I was unable to find the last two in the references (Thomas, & Van der Hoorn, 2018; Mukhi & al 2020). Anyway it seems little new information to me. The other referee also had a bunch of concerns, so I still can not recommend to publish the manuscript in Biology. Maybe a more specialized journal would be suitable.